# Oligonucleotide-Recognizing Topoisomerase Inhibitors (OTIs): Precision Gene Editors for Neurodegenerative Diseases?

**DOI:** 10.3390/ijms231911541

**Published:** 2022-09-29

**Authors:** Ben D. Bax, Dmitry Sutormin, Neil Q. McDonald, Glenn A. Burley, Tatyana Shelkovnikova

**Affiliations:** 1Medicines Discovery Institute, Cardiff University, Cardiff CF10 3AT, UK; 2Skolkovo Institute of Science and Technology, Moscow 121205, Russia; 3Signalling and Structural Biology Laboratory, The Francis Crick Institute, London NW1 1AT, UK; 4Department of Biological Sciences, Institute of Structural and Molecular Biology, Birkbeck College, London WC1E 7HX, UK; 5Department of Pure and Applied Chemistry & the Strathclyde Center for Molecular Bioscience, University of Strathclyde, Glasgow G1 1XL, UK; 6SITraN, Department of Neuroscience, University of Sheffield, Sheffield S10 2HQ, UK

**Keywords:** topoisomerases, inhibitors, gene editing, etoposide, camptothecin, CRISPR/Cas9

## Abstract

Topoisomerases are essential enzymes that recognize and modify the topology of DNA to allow DNA replication and transcription to take place. Topoisomerases are divided into type I topoisomerases, that cleave one DNA strand to modify DNA topology, and type II, that cleave both DNA strands. Topoisomerases normally rapidly religate cleaved-DNA once the topology has been modified. Topoisomerases do not recognize specific DNA sequences, but actively cleave positively supercoiled DNA ahead of transcription bubbles or replication forks, and negative supercoils (or precatenanes) behind, thus allowing the unwinding of the DNA-helix to proceed (during both transcription and replication). Drugs that stabilize DNA-cleavage complexes with topoisomerases produce cytotoxic DNA damage and kill fast-dividing cells; they are widely used in cancer chemotherapy. Oligonucleotide-recognizing topoisomerase inhibitors (OTIs) have given drugs that stabilize DNA-cleavage complexes specificity by linking them to either: (i) DNA duplex recognizing triplex forming oligonucleotide (TFO-OTIs) or DNA duplex recognizing pyrrole-imidazole-polyamides (PIP-OTIs) (ii) or by conventional Watson–Crick base pairing (WC-OTIs). This converts compounds from indiscriminate DNA-damaging drugs to highly specific targeted DNA-cleaving OTIs. Herein we propose simple strategies to enable DNA-duplex strand invasion of WC-OTIs giving strand-invading SI-OTIs. This will make SI-OTIs similar to the guide RNAs of CRISPR/Cas9 nuclease bacterial immune systems. However, an important difference between OTIs and CRISPR/Cas9, is that OTIs do not require the introduction of foreign proteins into cells. Recent successful oligonucleotide therapeutics for neurodegenerative diseases suggest that OTIs can be developed to be highly specific gene editing agents for DNA lesions that cause neurodegenerative diseases.

## 1. Introduction

This review concerns the potential use of oligonucleotide-recognizing topoisomerase inhibitors (OTIs) as gene editors (abbreviations used in this paper are listed at the end of the paper). We define OTIs as compounds in which a DNA-cleavage stabilizing topoisomerase inhibitor has been linked to an oligonucleotide-recognizing element. Such bifunctional OTIs are directed to cleave DNA at specific sites. OTIs are in many ways analogous to the guide RNAs in CRISPR/Cas systems, but rather than using an exogenous Cas protein to cleave DNA, they use endogenous DNA-cleaving enzymes–topoisomerases that are present in every active cell (we define active cells as cells which are transcribing DNA into RNA and/or replicating DNA). Because of this reliance on the intracellular cleavage machinery, OTIs have an advantage over CRISPR/Cas in gene editing applications in vivo (in patient) such as editing of disease-causative mutations in humans. The possibility of long-lasting correction by OTI gene editing, along with the progress on delivering oligonucleotides to the CNS, have prompted this review. We propose that OTIs may become an attractive strategy for monogenic diseases where a precise gene edit could potentially give a permanent correction. The spontaneous deamidation of cytosine (or 5-methyl cytosine) to uracil (or thymine) can, if not corrected by DNA-damage repair (DDR) processes, lead to permanent GC to AT transitions [1]. Some monogenic human diseases might be cured by reversing such mutations; for example some subtypes of amyotrophic lateral sclerosis (ALS) and potentially spinal muscular atrophy (SMA).

## 2. Gene Therapies for Neurodegenerative Diseases: Achieving “Permanent” Corrections

Today there are about 50 gene therapy clinical trials, a mix of antisense oligonucleotide (ASO) and virus-mediated transgene delivery based one, ongoing for Alzheimer’s disease (AD), Parkinson’s disease (PD), Huntingdon’s disease (HD), SMA and ALS [2]. ASO therapeutics typically target the underlying disease cause by modulating gene expression; they do not change the genome so most of them require repeated injections for a long-lasting effect [3]. On the other hand, viral-based delivery of human genes and associated regulatory elements, to replace or silence the expression of a faulty gene, ultimately promise “permanent” corrections where a single dose of an agent could be sufficient to correct the disease etiology.

Perhaps the most notable examples of transient and permanent gene correction are the two disease-modifying therapies recently approved for SMA. In December 2016 the US Food and Drug Administration (the FDA) approved the ASO drug nusinersen (Spinraza) for type I SMA; clinical studies showed dramatic improvement in infants [4,5]. SMA is caused by the loss of function of SMN protein due to a mutation or deletion in the *SMN1* gene, which if untreated can be lethal before the age of two [6]. However, the human genome has a second gene, *SMN2*, which encodes the same protein sequence, but which has a single silent point mutation causing a splicing variant such that the SMN protein encoded by *SMN2* normally lacks the exon 7 encoded amino acids [7]. Nusinersen promotes alternative RNA splicing of *SMN2* gene and thus increases production of functional SMN protein; it was developed by Ionis Pharmaceuticals-a company that specializes in developing antisense therapeutics. Nusinersen is an 18mer oligonucleotide in which the phosphates of the RNA backbone have been replaced by phosphorothioates and the 2’OH on the ribose groups have been replaced by a 2’-O-methoxyethyl [3]. Since oligonucleotides do not readily cross the blood–brain barrier, nusinersen is delivered every four months by direct injection (via lumbar puncture) into the cerebrospinal fluid [8]. An alternative transgene-based SMA treatment, onasemnogene abeparvovec (Zolgensma), was approved by FDA in 2019. The onasemnogene abeparvovec formulation contains the *SMN1* gene along with synthetic promoters encoded by a nonintegrating stable extranuclear episome [9] that is delivered to the patient via an adeno-associated virus serotype 9 (AAV9) [10]. Although the concept of permanent correction is extremely attractive, side effects from the delivery vectors remain the most significant caveat. Adeno-associated virus (AAV) vectors are amongst the most successful and popular gene therapy delivery methods [2,11]. However, AAV vectors can lead to immune responses and high doses have occasionally been lethal in clinical trials [2].

SMA represents the best known example of a monogenic neurological disorder that can now be treated by gene therapy. However, most adult-onset neurodegenerative diseases considered monogenic currently lack effective treatments, and ALS is one of the the most prominent examples. ALS is an extremely heterogeneous disorder, with >25 genetic subtypes identified for the familial disease (fALS) with the majority of mutations being missense mutations [12]. Genes most commonly affected by single amino acid substitutions are *SOD1*, *TARDBP*, *FUS* and *TBK1*, collectively accounting for ~40% of familial and ~10% sporadic cases and jointly bearing >250 different point mutations. The vast majority of these mutations alter the protein’s distribution and properties in a way that is not consistent with a clear loss of function or gain function mechanism [13]. One prototypical example is the *FUS* gene. Over 20 missense mutations were identified in its nuclear localization signal leading to cytoplasmic mislocalisation of the protein and causing both loss of its nuclear functions (e.g., in splicing) [14] and its cytoplasmic toxicity via aggregation [15]. Similarly, cellular pathomechanisms are yet to be established for ~50 mutations affecting TDP-43 protein encoded by the *TARDBP* gene [16]. Even mutations shown to be primarily causing gain of function mechanisms, e.g., in *SOD1*, also lead to at least partial loss of functionality [13]. Importantly, FUS, TDP-43 and other ALS-linked RNA-binding proteins regulate their own levels (autoregulation) potentially creating obstacles for changing the gene dosage; in addition, this may require introduction of bulky regulatory sequences such as introns [17]. Therefore, many ALS subtypes would require precision correction gene therapies, where the use of an mRNA degrading (splice-switching) ASO or a replacement gene might be of limited utility. Given the large spectrum of ALS mutations, access to a tunable therapeutic scaffold that can be easily personalized is also highly desirable.

An ideal therapeutic agent in the above cases might be a highly specific nucleic acid capable of precise gene editing that does not require exogenous enzymatic machinery, does not elicit an immune response, can be rapidly tuned and can be directly delivered into the CNS. In this review we examine the possibility that OTIs could become such agents-initially ‘correcting’ diseases caused by GC to AT transitions; applicable to some ALS cases and potentially to SMA.

## 3. DNA Topoisomerases, Anti-Cancer Drugs and OTIs

The double helical nature of DNA causes the accumulation of positive supercoils ahead of transcription bubbles and replication complexes and negative supercoils (Figure 1) or precatenanes behind [18,19]. Topoisomerases are essential enzymes that can modify the topology of DNA by creating temporary DNA-strand breaks in one (type I) or both (type II) DNA-strands [18,20,21]. Topoisomerases are needed to relax the positive supercoils that accumulate ahead of transcription bubbles and replication forks and to remove negative supercoils and precatenanes and catenanes that would otherwise accumulate behind transcription bubbles and replication forks [18,19,21,22]. Positive stranded RNA-viruses, such as SARS-CoV-2, require a specific type IA topoisomerase (Appendix A) for efficient replication [23]. Eukaryotic type IIB-like topoisomerases (Spo11 and Top6BL proteins in human, which cleave but do not religate DNA) are involved in formation of double-strand DNA breaks in meiosis [24,25,26], facilitating DNA exchange in sexual reproduction, and will not be considered further in this article. Note that topoisomerases are named such that odd numbered topoisomerases are type I (such as Top3A and Top3B-see Table 1), whereas even numbered topoisomerases are type II (such as topoisomerase IV and topoisomerase VIII-see Appendix A).

Table 1 gives an overview of human topoisomerases [29], topoisomerase targeting approved anticancer drugs [30] and derived OTIs [31,32,33,34,35,36,37,38,39,40,41,42,43,44]. Interestingly, although Top1 modifies DNA by creating single stranded DNA-breaks, camptothecins (Table 1) are only cytotoxic to cells in S-phase. In cells synthesizing DNA, the replication fork is believed to collide with ‘‘trapped’’ Top1-DNA complexes, resulting in double-strand breaks and apoptotic cell death [45,46,47]. A small percentage of patients treated with double-stranded break causing type IIA topoisomerase inhibitors have developed therapy related leukemias [48,49,50,51]. These type IIA topoisomerase therapy-related leukemias are due to balanced chromosomal translocations [51] in which, for example, the *PML* and *RARA* genes are rearranged to produce an oncogenic fusion protein [48]. These seem to be caused by two type IIA topoisomerases complexes producing spatially adjacent double-stranded DNA-breaks which are mis-repaired.

Oligonucleotide topoisomerase inhibitors (OTIs) described in the literature have been made by covalently linking drugs that stabilize DNA-cleavage complexes with topoisomerases either: (i) to DNA duplex recognizing triplex forming oligonucleotides (TFO-OTIs) or DNA duplex recognizing pyrrole-imidazole-polyamides (PIP-OTIs) (ii) or to oligonucleotides with sequences which form Watson–Crick base-pairs with a target DNA sequence (see Table 1 for references). Watson–Crick-OTIs have not yet been shown to strand invade a DNA duplex; this paper suggests strategies for devel-oping such WC-OTI strand-invasion.

### 3.1. Human Top1 and Top2α (/Top2β) Recognize the Phosphates on the DNA Backbone

The OTIs synthesized to date (Table 1) target either human Top1 or human Top2α and Top2β (which share a 68% amino acid sequence ID [52]). In human top2β [53,54] or top2α [55,56,57] crystal structures with DNA most protein contacts are with the phosphate backbone of the DNA. Similarly in binary complexes of Top1 with DNA [58,59], or ternary complexes with Top1, DNA and topotecan (or camptothecin) [45,46] most interactions are with the phosphate backbone (Figure 2). This is consistent with the activity of human Top1 and Top2α/2β being largely governed by DNA topology rather than specific base recognition [19]. 

In the simplified view of the function of Top1, shown schematically in Figure 2a, Top1 is shown having cleaved the top DNA-strand (based on Figure 4 panel G in [59]). The DNA is then believed to be relaxed by a controlled rotation, rotating about the phosphate group between nucleotides −1 and +1 on the uncleaved/intact strand [45,58]. The activity of Top1 is governed by the topological state of the DNA (Figure 1). Structures show a large number of interactions with phosphates on the ‘upstream’ (twenty red arrows in Figure 2a) side of the DNA-cleavage site. Inhibitors such as topotecan (Figure 2b purple) seem to block rotation by occupying the same space as the +1 base-pair.

### 3.2. Camptothecin Derived TFO-OTIs Target Type IB Topoisomerases

A 1997 study [31] showed that a camptothecin analog could give sequence specific DNA-cleavage by tethering it to a TFO. Camptothecin derived TFO-OTIs can now be modelled into crystal structures [58,59] of type IB topoisomerases with camptothecins [45,46] (Appendix A). TFOs bind in the major groove of a DNA-duplex, and tend to recognize runs of purines on one strand of the DNA duplex. In contrast pyrrole-imidazole-polyamides (PIPs) bind in the minor groove of a DNA-duplex, can recognize any sequence and have been used to make oligonucleotide-recognizing camptothecin derivatives [40,44] (see Appendix A for modelling of a PIP-OTI). Although camptothecin-TFO conjugates have been shown to be effective in targeting specific DNA sequences in cells [35,42], camptothecins (and OTIs based on camptothecins) seem to have quite a strong sequence preference at the DNA-cleavage site [60]. By convention DNA-cleavage sites cut by topoisomerases are numbered from −3, −2, −1 on the 5’ side and +1 +2, +3 on the 3’ side (the DNA is cleaved between nucleotides −1 and +1; there is no nucleotide with the number 0). Jaxel et al., 1991 [60] reported that 100% of 44 DNA-cleavage sites cleaved in the presence of camptothecin had a T at the −1 position and 75% had a G at +1 (see Figure 2b). The sites cleaved by a camptothecin based PIP-OTI [44], were consistent with this strong T G preference.

Etoposide (a type IIA inhibitor) based TFO-OTIs have also been described in the literature, with quite a long linker between the end of the oligonucleotide and the etoposide [34]. However, the authors stated that the linker arm used to conjugate the VP16 analog to the TFO was not long enough to span the number of nucleotides between the cut site and the TFO, raising the intriguing possibility that the TFO-etoposide might be bound to the T-segment (Figure 3a and Appendix A).

### 3.3. Type IIA Topoisomerases and Their Inhibition by the Anti-Cancer Drug Etoposide

Humans have two very similar type IIA topoisomerases, Top2α and Top2β. These two human type IIA topoisomerases are targeted by many anti-cancer drugs (Table 1) including anthracyclines, such as doxorubicin and daunomycin, anthracendiones such as mitoxantrone and epipodophyllotoxins such as etoposide [30,61]. Top2α plays the major role in DNA-replication while Top2β is expressed widely in post-mitotic cells where it is involved in transcription. Inhibition of Top2β by anthracyclines, such as doxorubicin and daunomycin, is thought to be responsible for cardiotoxicity that limits the dose of these drugs [61]. Top2α is believed to be the main target for anti-cancer drugs [61].

Etoposide is not a planar DNA-intercalator and stabilizes both single and double-stranded DNA breaks with human topoisomerases [62] and as well as with the bacterial type IIA topoisomerase DNA gyrase [63]. Figure 3a shows a generic type IIA mechanism. Schematics are shown based on: a 2.16Å DNA-cleavage complex of etoposide with human hTOP2β^core^ structure (Figure 3b) [54], and two structures of *S. aureus* gyrase^CORE^ fusion truncate [64] containing either two etoposide (Figure 3d) or one etoposide (Figure 3e). In these crystal structures etoposide sits in the DNA-cleavage sites physically preventing DNA-religation. The complex with only one etoposide bound has a larger area buried between the two subunits at the DNA-gate-suggesting the DNA-gate is more closed in the complex with one etoposide than in that with two etoposides [63].

### 3.4. Sequence-Selective DNA Cleavage Using First-Generation Watson–Crick-OTIs

The structures shown in Figure 3b,e were used in modelling OTIs with a single etoposide covalently attached to an oligonucleotide [43]. These Watson–Crick type-OTIs were able to specifically cleave a DNA strand whose sequence was that of an oncogenic PML-RARA breakpoint fusion [43] (Figure 3). The OTIs designed by Infante Lara et al., (2018) aimed to produce single stranded cleavage in the target sequence and did so (Figure 4). In particular it was shown that a 30 mer OTI cleaved a target oncogenic fusion DNA sequence with high specificity (although a 20 mer did not cleave so well) [43]. Interestingly, one of the sites cleaved with an OTI and human Top2α had hardly any cleavage in the presence of 500 mM etoposide [43], suggesting WC-OTIs might be able to target any DNA sequence.

Figure 4 shows two OTIs, a 50 mer and a 30 mer, which are the same apart from the length of the OTI, and both cleave the target oncogenic DNA target sequence more effectively than 500 mM etoposide (see Figure 9 in [43]). The assay used detected the cleavage of the DNA target sequence by radio-labelling the 5’ end of the target oligonucleotide. Although Infante Lara et al. [43] showed that by coupling the drug etoposide to an oligonucleotide the OTI could target DNA-cleavage of a specific complementary DNA strand, DNA strand-invasion (i.e., DNA-cleavage of a plasmid or a DNA duplex containing the target nucleotide sequence) has not yet been demonstrated [43]. Watson–Crick type OTIs have not yet been reported for Top1 targeting compounds.

### 3.5. Achieving Strand Invasion for Watson–Crick-OTIs

In a CRISPR/Cas system, in order to start ‘melting’ the DNA-duplex, to allow strand invasion of the guide-RNA to take place, the Cas protein recognizes the three nucleotides of the protospacer adjacent motif (PAM) [65]. Once the DNA duplex has started to melt at the PAM motif the RNA can strand-invade. RNA-DNA duplexes are often more stable than DNA-homoduplexes. A functional equivalent to the PAM type motif, to allow strand invasion of WC-OTIs to take place, is yet to be developed. One strategy to achieve DNA strand-invasion could be cleavage of the single DNA strand that the OTI is aiming to replace. Such a scheme is shown in Figure 5, with initial single-stranded cleavage being accomplished by an artificial metallonuclease (AMN) moiety attached to a TFO region (cyan). This TFO-AMN (cyan) region of the oligonucleotide is envisioned to produce multiple single-stranded nicks in cellular DNA, but strand-invasion by the WC-OTI region (orange) is only envisioned as taking place when a complementary region is adjacent to the single-stranded DNA-cut site (Figure 5c).

Two recent papers have suggested that DNA-targeted metallodrugs may become suitable agents for gene editing by themselves [66,67]. In DNA-targeting metallodrugs, a metal chelating chemical moiety is covalently attached to an oligonucleotide to cleave DNA at a particular position [66,67]. In a 2020 paper describing this chemistry-based approach, an artificial metallonuclease (AMN) that oxidatively cuts DNA, was coupled to a TFO, to cleave specific DNA sequences without any enzyme [67]. This TFO-AMN approach is somewhat reminiscent of the camptothecin-TFOs, which target type IB topoisomerases to cleave a specific DNA sequence. The developed TFO-AMNs suggest an approach for getting a Watson–Crick-OTI to strand-invade, as shown in Figure 5. Other approaches to encourage WC-OTI strand-invasion could also be devised.

## 4. Using OTIs to Exploit ‘Safe’ DNA Repair Pathways in the CNS

DNA-cleavage stabilizing anti-cancer drugs (Table 1) kill cancer cells by creating multiple double-stranded DNA-breaks. The bacterial immune CRISPR/Cas systems cuts up (makes double stranded DNA-breaks in) the DNA of invading bacteriophage. However, while making double stranded DNA-breaks for gene editing has advantages, it is potentially hazardous and therapy related leukemias have been reported in the literature [10,29,55,56,69]. For this reason, we suggest utilizing single stranded breaks for neurodegenerative in vivo gene editing efforts with OTIs.

Chatterjee and Walker, reviewing DNA damage, repair and mutagenesis [1], described how spontaneous deamination of 5-methyl cytosine produces thymine (Figure 6). The resulting G-T base-pair is recognized by thymine DNA-glycosylase and repaired. Apparently, GC to AT transitions account for at least one third of single-site mutations responsible for hereditary diseases in humans [1], therefore ‘correcting’ such GC to AT mutations seems to be a reasonable initial target for OTI therapeutics. Although the 5′-flanking base pair to G·T mismatches influences the rate of removal of thymine [70] we assume, for the sake of simplicity in this review, that the G-T mismatches will eventually be repaired in cells in the CNS. Further experiments will need to be performed to demonstrate this; however, other DNA repair pathways [1] might be exploited for ‘safe’ gene editing in the CNS.

## 5. OTIs: Better In Vivo (In-Patient) Gene Editors Than CRISPR/Cas?

Engineered nucleases called zinc finger nucleases (ZFN), transcription activator-like effectors nucleases (TALENs) and the CRISPR/Cas9 system are the most well-known DNA-targeting gene editing systems [66,71]. All three, once delivered into prokaryotic or mammalian cells, can create double-stranded breaks at desired genomic loci [66,72]. The CRISPR/Cas9 system has revolutionized genomic editing [73,74] because it lacks inherent limitations of the other two systems, e.g., does not require complex design/assembly steps. Although immensely useful in the research and for ex vivo gene editing [75], where the delivery can be achieved with relative ease, the use of CRISPR/Cas in the context of human patients is associated with a number of caveats.

### 5.1. Delivery

Efficient delivery of all the editing components to the site of action still remains a major obstacle in the use of CRISPR/Cas. A minimum of two components have to be delivered into cells for CRISPR/Cas9 based target cleavage, the sgRNA and the Cas9 enzyme. To be able to carry out precise targeting, a repair template is also required. Although Cas9 does not have to be delivered as a ready enzyme (protein), and the Cas9-encoding DNA (ORF) or mRNA can be delivered instead, sometimes even fused to sgRNA, the delivery of such a large nucleic acid species remains challenging. Lentiviruses, adenoviruses and AAVs were tested as delivery vectors, with AAVs being most popular in gene therapy clinical trials due to a number of favorable characteristics [76]. However, AAV vectors have a limited packaging capacity (~4.7 kb), whereas Cas9 alone has a genetic size of ~4.5 kb, and together with sgRNA (s), the total plasmid size can exceed 7 kb, making it impossible to deliver all components using one vector. In contrast, delivery of OTIs, similar to other therapeutic oligonucleotides, due to their small size can be achieved with relative ease both locally and systemically, for example, intrathecal infusion [77]. Non-viral delivery methods include lipid and polymer-based nanocarriers such as nanoparticles/liposomes, gold nanoparticles or inorganic nanoparticles [78,79].

### 5.2. Immune Reactions to ‘Foreign’ Proteins

Recent findings show that the introduction of Cas9 into the human body may result in immunogenicity. Antibodies to the two most widely used orthologs of Cas9, SaCas9 and SpCas9, alongside anti-SaCas9/SpCas9 T cells were identified in over 50% of serum donors [80,81]. Whilst not being of much concern when used for ex vivo therapies, these preexisting humoral and cell-mediated adaptive immune responses can elicit cytotoxic T cell response specifically against Cas9-expressing cells. On the other hand, OTIs and DNA-targeted metallodrugs [66,67] do not require introduction of foreign proteins and are free of this inherent problem of CRISPR/Cas9.

### 5.3. Sequence-Specific Targeting and Strand-Invasion

Sequence-specific targeting [82,83] is relatively well developed for CRIPSR/Cas systems where the sgRNA has to stand-invade the target DNA-duplex. The recognition starts at an adjacent PAM sequence required to initiate DNA melting [84,85,86]. Kolesnik et al. [87] suggested preliminary PAM recognition may reduce the number of sites in a genome to be fully melted and screened, thus accelerating the process of target searching. For OTIs experiments need to be done to investigate sequence-specific targeting (and strand-invasion for WC-OTIs). Spontaneous in vivo formation of RNA-DNA heteroduplexes (R-loops) is well known in prokaryotic and eukaryotic cells [88]. The R-loop model may be used as a template for target recognition by a DNA-duplex invading WC-OTI. R-loops can efficiently nucleate at G-rich clusters [89] and are favored by excessive negative supercoiling that destabilizes the DNA duplex [90,91]. Thus strand invading WC-OTIs could be designed to recognize G-rich clusters to initiate strand invasion and then use Top2β for cleavage (Top2β is involved in relaxation of transcription-induced negative supercoiling [92]). 

## 6. OTIs: A Tunable Scaffold for the Correction of Amyotrophic Lateral Sclerosis Causative Mutations?

As stated earlier, a large population of ALS patients could potentially benefit from the use of OTIs as gene editors. Here, a single ALS-causative mutation in the *SOD1* gene, G37R, successfully used for disease modelling in mice [93], is used as a case study. Three camptothecin based PIP-OTIs are used for (‘a three cuts and you are out’–strategy to correct this mutation [94] as shown in Figure 7 (see also Appendix A). In Figure 7, three PIP-OTIs are envisioned as producing three DNA-cleavage sites. DNA-repair is then envisioned to remove the ‘cyan’ and ‘orange’ cleaved regions of one DNA-strand (Figure 7b) before this replacing with an oligo (Figure 7c). Experimental cellular verification of this (or other strategies) is needed (see also discussion).

Strong DNA-sequence specificity was demonstrated for camptothecin based DNA-cleavage [60], and experiments to date suggest this extends to camptothecin PIP-OTIs [44]. This suggests camptothecin-based OTIs primarily produce a single stranded DNA-cleavage between at a T-G sequence on the cleaved strand. A similar strategy could potentially be applied for some fALS mutations in the *FUS* gene, e.g., R521H (*G1562A*) and R518K (*G1533A*) [95].

## 7. OTIs: Could a Single Nucleotide Edit Cure Spinal Muscular Atrophy?

Humans have two genes, *SMN1* and *SMN2*, that encode identical protein sequences [6,7], whereas other mammals including primates have only one gene [6]. The loss of the *SMN1* gene causes SMA because the paralogous *SMN2* gene is differentially spliced (~90% of the time) and the resultant mRNA normally lacks exon 7 [6,7], the amino acids encoded by exon 7 are critical for protein oligomerisation and function [6]. A single silent point mutation, at position 6 of exon 7, gives an exon splicing enhancer in *SMN1* but an exon splicing silencer in *SMN2* [6]. In human evolution, the *SMN1* gene is believed to have been duplicated to give rise to the *SMN2* gene (or *SMN2* genes;-different people have different numbers of *SMN2* genes) [6]. A single GC to AT transition at position 840 (exon 7) is responsible for the different splicing of *SMN2* [7], therefore changing this AT base-pair in the *SMN2* gene into GC will convert it into *SMN1* thereby theoretically curing SMA (Figure 8). Figure 8 shows an editing scheme similar to that in Figure 7, but with a different target sequence. However, complications may arise due to the existence of repetitive elements in the *SMN2* genomic region; which make it prone to rearrangements and deletions. Therefore, careful cellular and in vivo [96] validation studies would be required to establish a proof-of-principle. Encouragingly, CRISPR-based *SMN2* gene conversion was achieved in human iPSCs and rescued SMN protein levels [97].

## 8. Discussion: Can OTIs Combine the Powers of CRISPR/Cas and ASOs and Lack Their Inherent Weaknesses as In Vivo Gene Editors?

The development of the ASO therapeutic nusinersen, showed how by modifying the phosphate backbone with phosphorothioates and using a 2’-O-methoxyethyl [3] a long lasting stable (resistant to nucleases) therapeutic agent could be made. However, for OTIs, although such modifications can be modelled on the computer graphics, it is difficult to predict affinities and specificities. So a ‘hit-to-lead’ optimization cycle for OTIs might contain four stages:modelling nuclease resistant and OTI-target stabilizing mutations using computer graphics (and existing crystal structure from the PDB).chemical synthesis of about ten such OTIs (PIP-OTIs, and/or TFO-OTIs/WC-OTIs).biophysical assays (such as swithSENSE-[98]) to optimize relative affinities of modified OTIs for target DNA sequences ‘in vitro’-with purified topoisomerases.using nuclease resistant OTIs in iPSC cells to compare their gene editing functionality with that of published CRISPR/Cas systems for similar diseases (e.g., [97,99,100]). We assume initially that OTI development pathways would aim at replicating and improving on existing CRISPR/Cas gene editing in iPSCs. How many optimization cycles it would take to achieve such an aim for a particular mutation is not yet clear.

In this review, an outline of how OTIs could be directed to cleave a particular DNA sequence [31] is proposed. Because so much of drug development concerns safety, the initial OTIs we have proposed in this review focus on cleaving only one DNA-strand (Figure 5, Figure 7 and Figure 8). Note that the fate of OTI stabilized DNA-cleavage complexes in a cell will depend on the DNA repair mechanisms active in that particular cell type [101]. Here, we suggest that, for the post-mitotic cells of the CNS, thymine DNA glycosylase should naturally repair G-T mismatches created by single strand OTI-based gene editing, in a relatively safe manner. Replicating ex vivo cell-based CRISPR/Cas9 gene editing experiments with OTIs may be a way forward to establish a proof of principle for their in vivo (in patient) development. For adult-onset monogenic neurodegenerative diseases correction after the disease onset still may provide a clinical benefit [102], but early correction, after genetic testing, could become a cure.

Creating double stranded breaks in DNA, in a manner similar to that used by CRISPR/Cas systems when they cut up the DNA of invading bacteriophages, might be useful for OTIs targeting the cutting up of DNA encoding antimicrobial resistance genes in pathogenic bacteria. However, obtaining OTIs with specificity for bacterial over human topoisomerases and how to deliver such OTIs to bacteria may be challenging (see also supplementary discussion and Appendix A). DNA-cleavage stabilizing topoisomerase inhibitors (Table 1) are widely used in cancer chemotherapy [29,61,103] and potential uses of OTIs in cancer are discussed in the supplementary discussion. Exactly how eukaryotic topoisomerases interact with chromatin and other gene regulatory elements is still the subject of research [21], so although we suspect that WC-OTIs will be able to cleave anywhere in a transcribed gene, more experiments are needed to prove this. Using OTIs to create double stranded DNA breaks to cut-up genes encoding oncogenic fusion proteins may potentially be beneficial, but genetic heterogeneity and genomic instability are both hallmarks of many cancers [104], so how safely this would work in the clinic remains uncertain. Therapy related leukemias have been reported both for some gene therapies [69,105,106,107] and for topoisomerase II drugs [48,50,108,109]. So for gene therapy in vivo (in patient) Top1 targeting OTIs [31,38,39,40,41], that cleave only one DNA-strand, may be safer to develop. 

The recently reported therapeutic effect of the ASO Tofersen in some adult patients with SOD1 ALS, is a futher proof that ASOs can be successfully delivered and exert desired activity in the CNS [110]. However, better therapeutic options for patients with ALS are clearly needed [111]. The recent clinical approvals of oligonucleotide-based therapeutics [112], together with the remarkable promise of genomic sequencing based personalized medicines [113], suggests novel therapeutics, such as OTIs, that specifically target differences in the DNA sequences between normal and lesioned cells in patients will be worth developing.

## Figures and Tables

**Figure 1 ijms-23-11541-f001:**
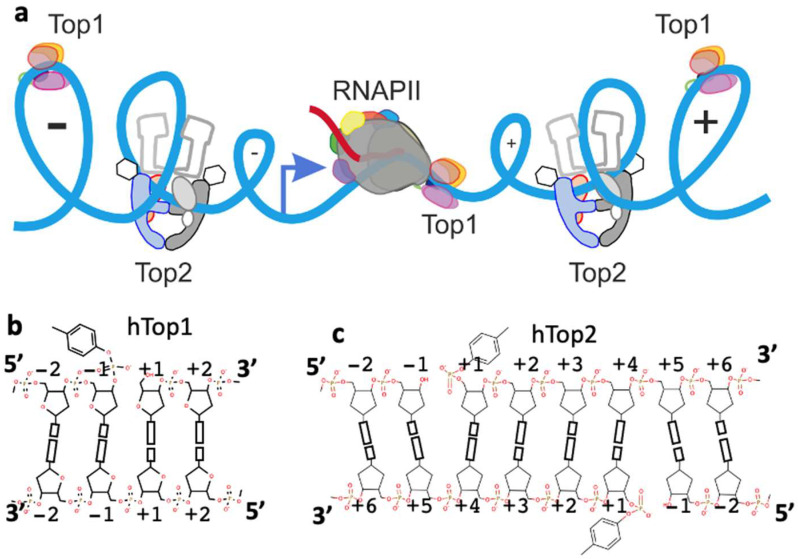
Human Top1 and Top2 regulate DNA topology at a transcription bubble. (**a**) In the post-mitotic cells of the central nervous system-human Top1 and human Top2β are expected to be active in regulating DNA topology near transcription bubbles. The only DNA replication in post-mitotic CNS cells will take place in mitochondria; and careful design may be needed to ensure that damage to replicating mitochondrial DNA does not take place [27]. RNAPII, RNA polymerase II, is shown in contact with Top1 [28]. (**b**) Human Top1 cleaves a single DNA strand forming a 3’ phosphotyrosine. By convention both cleaved and non-cleaved strands are numbered relative to the single DNA-cleavage site. (**c**) Human Top2 (Top2α or Top2β) can cleave both DNA strands forming two 5’ phosphotyrosines. By convention both strands are numbered relative to the two 4-base-pair staggered DNA-cleavage sites.

**Figure 2 ijms-23-11541-f002:**
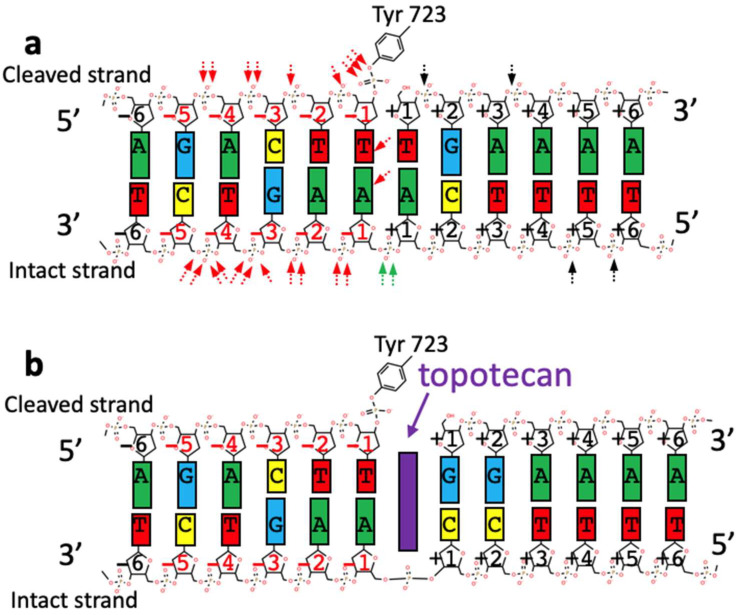
Simplified schematics of twelve base-pairs of DNA in TOP1 DNA-cleavage complexes. (**a**) Simplified schematic of the central twelve base-pairs of DNA in a Top1 DNA-cleavage complex (based on 2.1Å structure pdb code: 1a31). Interactions (<3.5Å) between the protein and the DNA are represented by arrows (adapted from Figure 4–, panel G in [59]). Tyrosine 723 from Top1 has cleaved the top strand. Top1 can be imagined as a hand holding the double-stranded DNA up-stream of the DNA-cleavage site (red arrows and–red numbered nucleotides) and allowing controlled rotation about the phosphodiester bond between the −1 and +1 nucleotides on the intact strand (two green arrows) to relax the DNA. (**b**) A simplified schematic of the same DNA sequence in a complex with topotecan (a camptothecin derivative). The figure is based on the 2.1Å structure with topotecan (pdb code 1K4T). Note the +1 G-C base-pair (in 1K4T). The topotecan occupies the ‘same’ space as the +1 nucleotide pair in panel a. ‘The intercalation binding site is created by conformational changes of the phosphodiester bond between the +1 and −1 base pairs of the intact strand’ [45].

**Figure 3 ijms-23-11541-f003:**
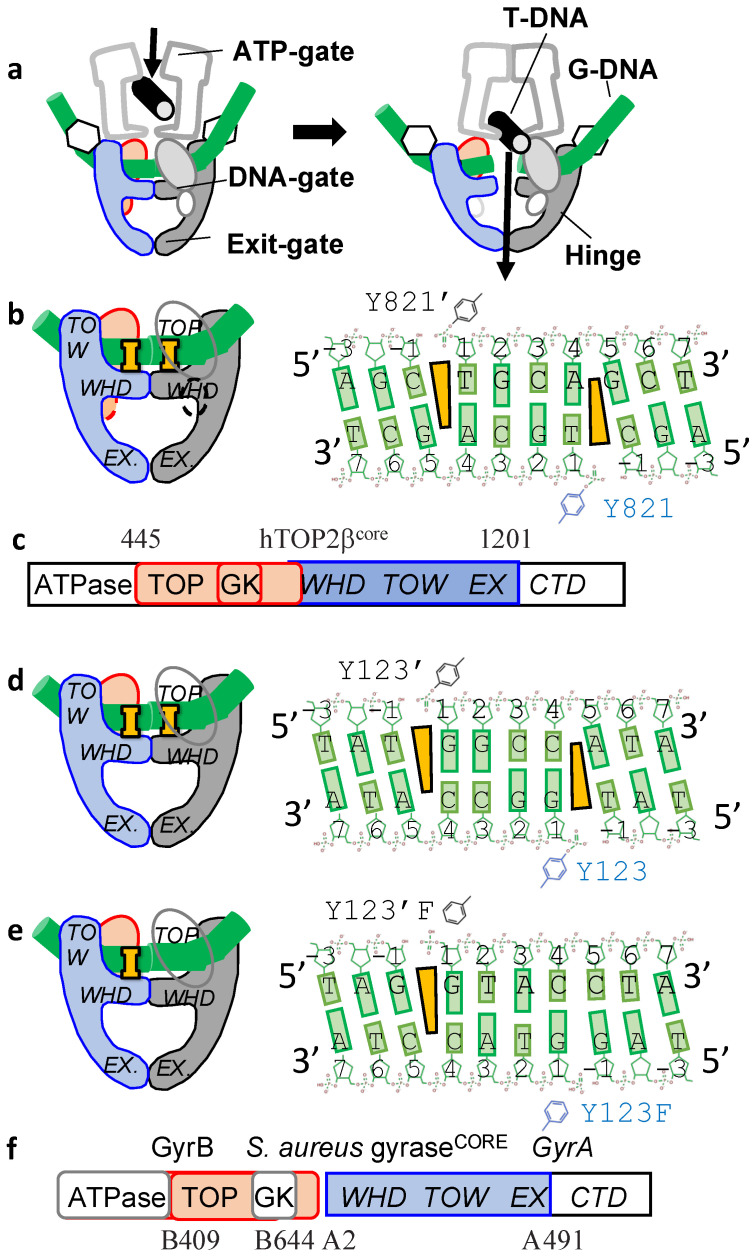
Type IIA topoisomerases and schematics of three etoposide crystal structures. (**a**) Simplified schematic of a reaction carried out by a type IIA topoisomerase. The gate or G-DNA (green cylinder) is cleaved and another DNA duplex, the T (or transport segment;-black) is passed through the cleaved DNA before religation. (**b**) Schematic of a 2.16Å human hTOP2β^core^ structure (pdb code: 3qx3) with DNA and two etoposides (I) binding at the two DNA cleavage sites, four base-pairs apart. One subunit is shown in red and blue, the other in grey. The DNA sequence (5′-3′) is the same for both strands; the DNA has been cleaved by tyrosine 821. (**c**) In human TOP2β is a single subunit and functions as a homodimer; the hTOP2β^core^ is residues 445-1201. Structural domains in hTOP2β^core^ are TOPRIM (TOP) domain, GK = Greek key domain, WHD = winged helical domain, TOW = tower domain, EX = exit gate domain. (**d**) Schematic of a 2.8Å structure of *S. aureus* gyrase^CORE^ fusion truncate DNA complex containing two etoposide (I) binding at the two DNA cleavage sites (pdb code: 5cdn), four base-pairs apart. One gyrase^CORE^ fusion truncate is shown in red and blue, the other in grey. (**e**) Schematic of a 2.45Å structure of *S. aureus* gyrase^CORE^ fusion truncate DNA complex containing one etoposide (pdb code: 5cdp) (I) (**f**) DNA gyrase consists of two subunits, GyrB and GyrA (domains are indicated). Note in the *S.aureus* gyrase^CORE^ fusion truncate the GyrB and GyrA subunits are fused into a single ‘subunit’ (B409-B644 + A2-A491) and the small greek key domain (residues B544-B579) has been deleted.

**Figure 4 ijms-23-11541-f004:**
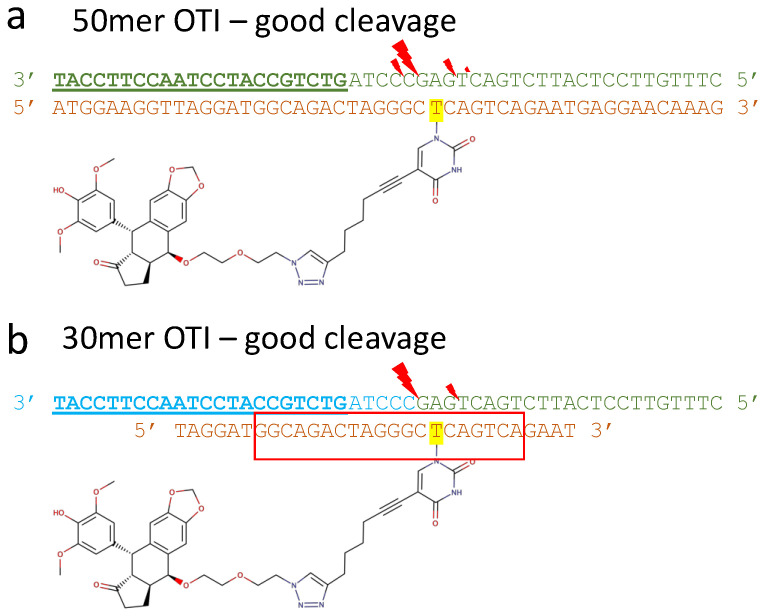
50 mer and 30 mer Watson–Crick-OTIs cleave an oncogene target sequence. Adapted from Figure 9 in [43] focusing on Top2α cleavage. (**a**) The target oncogenic *PML-RARA* sequence to be cleaved is shown in green on the top line, with the *RARA* sequence bold and underlined at the 3′ end; the 5′ sequence is from *PML*. The bottom line shows the 50 mer OTI with an etoposide moiety covalently attached to the T (highlighted in yellow). DNA-cleavage gels showed that the OTI promoted DNA-cleavage at four sites on the target (green) strand with relative intensities (in brackets) 24–25 (17), 23–24 (45), 20–21 (17), 19–20 (5). (**b**) The same target *PML-RARA* sequence is shown on the top line, but coloured in cyan at the 3′ end to indicate the position of the major cleavage site. Only two DNA-cleavage sites were observed with the 30 mer OTI, 23–24 (32), 20–21 (9) (red lightning bolts show positions). The red box indicates where the 20 nucleotides are from Xtal structures with etoposide (pdb codes: 3qx3, 5cdn and 5cdp);–based on the major DNA-cleavage site.

**Figure 5 ijms-23-11541-f005:**
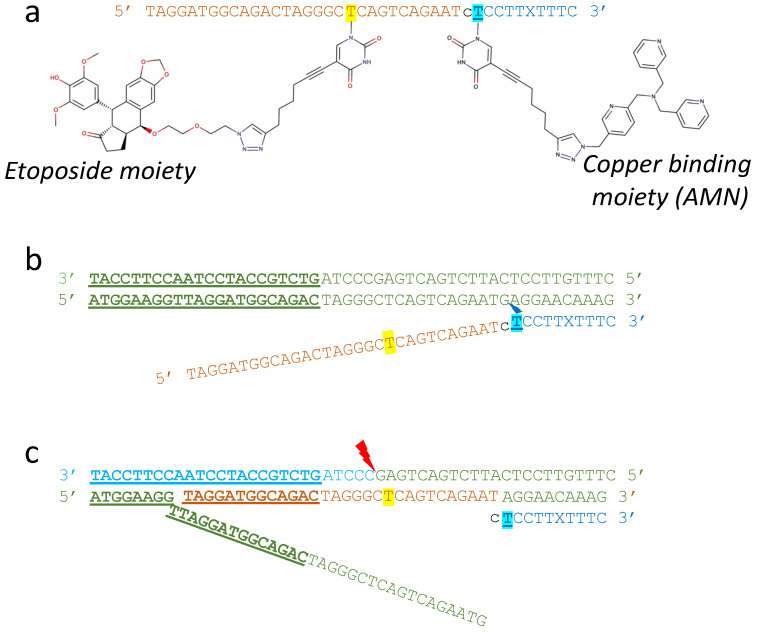
A 41 mer comprising a 30 mer etoposide-OTI coupled to a 10 mer AMN-TFO. (**a**) Depiction of a 41 mer, in which a 30-mer WC-OTI (orange letters with etoposide attached to yellow highlighted T) has been linked (single c in black) to a ten nucleotide TFO (blue letters) with a copper binding artificial metallo-nuclease (AMN) at the 5′ end. The copper binding AMN moiety is covalently attached to a cyan highlighted T. Note the X is a nucleotide designed to recognize a C in a TFO [68]. (**b**) the OTI recognizes a DNA-duplex (green letters) and the AMN cleaves one strand of the duplex (blue lightning bolt). (**c**) The rest of the strand-invading OTI (orange letters) can now strand invade and cleave the target oncogene (red lightning bolt–as in Figure 4b).

**Figure 6 ijms-23-11541-f006:**
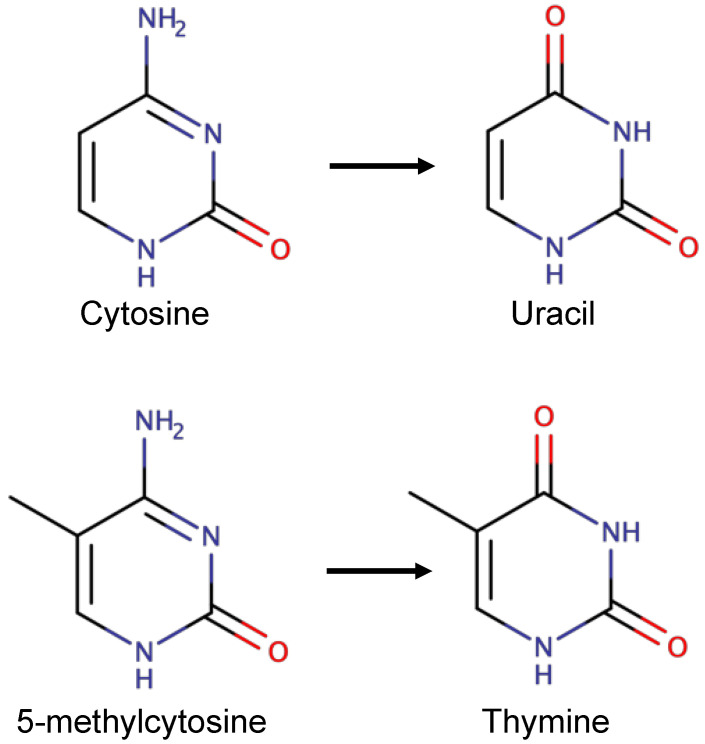
Deamidation of cytosine to uracil and 5-methylcytosine to thymine. The exocyclic amine of cytosine can be spontaneously deamidated to give uracil and the excocyclic nitrogen of 5-methylcytosine can be spontaneously deamidated to give thymine. Uracil is removed from DNA by uracil-DNA glycosylase while the G:T base-pair resulting from spontaneous deamidation of 5-methylcytosine can be removed by thymine DNA glycosylase (Figure drawn with Marvin-Sketch, from ChemAxon, https://www.chemaxon.com).

**Figure 7 ijms-23-11541-f007:**
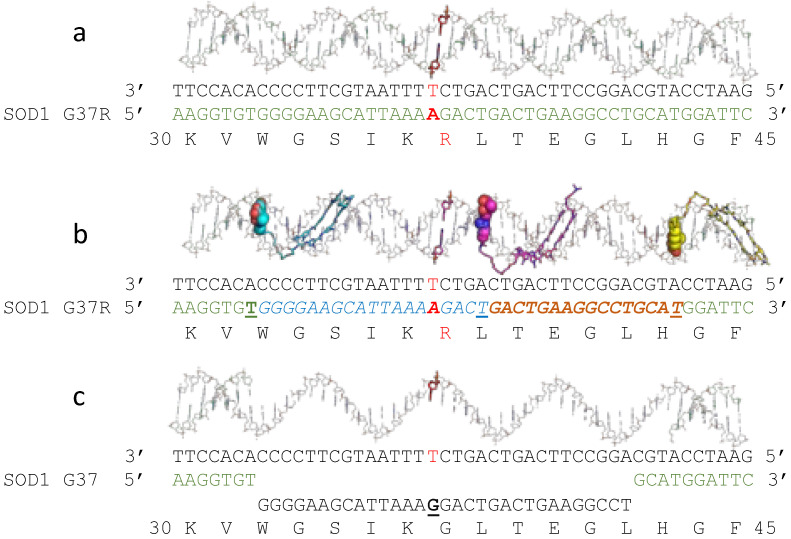
Modelling a theoretical correction by three camptothecin based PIP-OTIs of a mutation associated with familial ALS. (**a**) A model (BDNA, 8 June 2022, from http://www.scfbio-iitd.res.in/software/drugdesign/bdna.jsp) of nucleotides (one strand carbons in black, the other in green) coding amino acids 30–45 (single letter code bottom line) of human *SOD1* with a G37R mutation. The single base-pair change causing the glycine to arginine mutation is highlighted in red. (**b**) Three camptothecin based PIP-OTIs are shown coloured with cyan, magenta and yellow carbons, with the intercalating camptothecin moiety in solid space-fill representation. Three positions (TG) cut on the lower strand, in the presence of Top1 (see also Appendix A) are indicated. In cleavage complexes the T is covalently bonded to Top1 by a 3′ phosphotyrosine bond and the camptothecin moiety intercalates between the TA and GC base-pairs at the DNA-cleavage site. (**c**) The blue and orange oligonucleotides in b are envisioned to have been removed – and are replaced with an oligonuleotide with a corrected G. The T in the central G-T mismatch should be removed by thymine DNA-deglycoylase–after which the red T in the top strand should be corrected to C.

**Figure 8 ijms-23-11541-f008:**
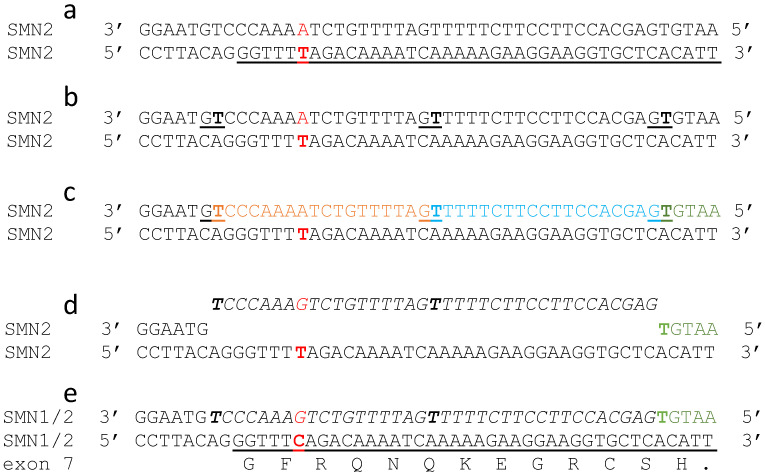
A theoretical correction, by three camptothecin based PIP-OTIs, of the exon 7 mutation associated with SMA. (**a**) DNA sequence from human *SMN2* gene (NCBI’s RefSeq gene ID: 6607). Experimental evidence suggests that Exon 7 (underlined sequence–bottom line) is skipped because of alternative splicing due to a single base change (AT base-pair highlighted in red). (**b**) Three positions (TG) positions (5′–3′) on the upper, non-coding strand, to be targeted for cleavage by PIP-OTIs are underlined (note upper strand is drawn 3′–5′). (**c**) In the presence of Top1 (see also Appendix A) three PIP-OTIs are predicted to cleave the DNA at three positions and remain covalently linked to the T’s. (**d**) After removal of the PIP-OTIs and covalently attached DNA–a gene editing oligonucleotide with a corrected ***G*** is introduced. (**e**) After the ‘theoretical’ correction of the G-T (in panel **d**) mismatch to G-C (panel **e**) exon 7 should be expressed (as in *SMN1*).

**Table 1 ijms-23-11541-t001:** Human topoisomerases, DNA-cleavage stabilizing anticancer drugs and derived OTIs.

Type of Topo Polarity Mechanism	*Gene Name*	Protein Name	Drug (Class) US Approval Date (Comments)	Type of OTI. Date First Publication (References)
IA	*TOP3A*	TOP3A	None yet	None
5’-PY	*TOP3B*	TOP3B		
Strand passage				
IB 3’-PY Rotation	*TOP1* *TOP1MT*	Top1 Top1mt	Topotecan (camptoth.) 1996 (mitochondrial Top1mt not specifically targeted)	Camptothecin-TFOs 1997 [31,32,33,34,35,36,37,38,39,41,42]. Camptothecin-PIPs 2001 [40,44]
IIA 5’-PY Strand passage /ATPase	*TOP2A* *TOP2B*	Top2α Top2β	Doxorubicin (anthracyc.) 1974 Etoposide (epipodophy.) 1983 Mitoxantrone (anthracen.) 1987	Daunomycin-TFOs * 2008 [33]. Etoposide-TFOs 2006 [34] Etoposide-Watson–Crick-OTIs ** 2018 [43]

TFO = triplex forming oligonucleotide, PIP = Pyrrole-imidazole polyamides, * Daunomycin is also know as Daunorubicin and is an anthracycline. ** Etoposide-Watson–Crick-OTIs cleave complementary DNA strands in vitro. Drug class abbreviations: camptoth. = camptothecin; anthracyc. = anthracycline; epipodophy. = epipodophyllotoxin; anthracen. = anthracenedione.

## Data Availability

Not applicable.

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
