# Peer review of "Oligonucleotide-Recognizing Topoisomerase Inhibitors (OTIs): Precision Gene Editors for Neurodegenerative Diseases?"

_ijms, 2022, doi:10.3390/ijms231911541_

Round 1

Reviewer 1 Report

Bax et al. provide a compelling argument for a new class of targeted oligonucleotide-recognizing topoisomerase inhibitors (OTIs) with the theoretical potential to cure monogenic diseases. This treatment would involve directing topoisomerase (TOP) poisons towards mutation sites enabling cutting of the DNA with endogenous TOPs, alongside transfection with an oligonucleotide allow the DNA repair machinery to effectively replace the harmful mutation with the wild-type residue. Throughout the review, the authors describe the current state of the gene therapy field, then proceed to describe the OTI mechanism, their advantages over other forms of gene therapy, and the author’s specific hypothesis for the future of OTI usage. While the underlying ideas are good, the manuscript is occasionally repetitive, confusing and lacking some key information pertinent to the success of OTI therapy going forward. I would recommend further revision before publication. My comments are outlined below:

Major points:

·      The structure of this manuscript is very strange, where the entire description of the mechanism of action for OTI activity is described in detail in the figure legend while barely referred to in the main text (e.g. Figs 5, 7 & 8). Considering there is so much extraneous and repetitive material in the main text (see following point), it would be much simpler to include at the very least a short mechanistic description for each Figure in the main text, then leave additional technical information in the legends. For a specific example, it is very jarring to have the colloquial term “a three cuts and you are out” be the only description of Figure 7 in the main text, despite it describing the key underlying mechanism of the whole review.

·      There are many parts of the review which are repetitive. For example, much of the same information about SMN1 and SMN2 is repeated in Sections 2 & 7. Also, the sequence similarity of Top2α and Top2β is repeated at the start of Sections 3.1 and 3.3. These are not the only examples of unnecessary repetition. The authors should go through the manuscript and remove similar instances. At the moment, it gives the impression of having been written in sections by different people.

·      For this to be a comprehensive outline of the OTI strategies, the review must highlight the potential pitfalls of the technology. The authors will have a much better understanding of the potential roadblocks to this technology than me, but at the very least, the following points should be considered:

o   What is the predicted half-life of OTIs? Are they any assumptions that can be made from other gene therapy?

o   TOP poisons are most commonly (perhaps in all cases?) reversible, such that reducing the drug concentration can allow the drug to leave the cleavage site and allow the TOP to religate the break and move away. In light of the above point, might it be unlikely that the “three cuts” strategy would work if it require all three TOPs to be engaged at once without moving away?

o   It is unclear to me why the cut ssDNA fragments would simply denature and disengage from the genomic DNA rather than stay annealed in standard cellular conditions?

o   The authors correctly mention the replication-dependent genotoxic effects TOP1 poisons. Presumably this will still be an issue even with targeted TOP poisoning activity. Could there be complications if the OTI cause replication-runoff DSBs at the sites of the very genes they are trying to edit?

·      Line 133: Spo11 and Top6BL are not topoisomerases, as they cannot religate the cleaved DNA. They are topoisomerase-like. Although this means they can in theory be harnessed with OTI systems, this must be corrected and in Table 1.

·      Line 478: “…for our lack of success in obtaining funding.” This is a very strange framing for the manuscript! I would definitely reword/remove this.

Minor points:

·      Line 29: “…to take place to give strand…” Confusing and unclear sentence

·      Line 39: In Abbreviations section, not Appendix A

·      Line 148: Why does Figure 1C specifically describe Top2β, when this mechanism is exactly the same at Top2α? Can be replaced and described at simply Top2.

·      Lines 209, 228 and 447: It is completely unnecessary to directly quote manuscripts when the key information can be reworded.

·      Line 245: It isn’t necessary to point out the resolution of the structure if it is simply a structure schematic.

·      Line 252: Figures 3C and 3F not referred to in the text. Are they relevant?

·      Line 259: In human, not “in man”

·      Lines 280-283: Reword the concept so it isn’t so specific to the experimental conditions of another paper.

·      Line 433: What does space-fill mean?

·      Line 485: “targeting”

·      Line 517: “simultaneously”

·      Line 521: Funding section has not been filled in

Reviewer 2 Report

The review is focusing on discussing the potential application of topoisomerase inhibitors on gene edition. It is an interesting proposal. However, less evidence has been shown that topoisomerase inhibitors have specific sequence preferences that can be used as guidance for gene therapy.  

It is worth seeking, but not reasonable without providing more solid reports and supports for this hypothesis. 

Reviewer 3 Report

In this review article Bax and Collogues have highlighted potential use of oligonucleotide-recognizing topoisomerase inhibitors (OTIs) as gene editors. Authors have presented an outline of how oligonucleotide-recognizing topoisomerase inhibitors (OTIs) can be directed to cleave a particular DNA sequence has been given. Authors have also discussed new strategies to enable DNA-duplex strand invasion of WC-OTIs to take place to give strand-invading SI-OTIs.

Recently, many Oligonucleotides are being investigated at clinical settings. In my view this is a timely review in the area of oligonucleotide-based therapeutics.

Authors have presented topics very well and have summarized key information concisely. I am in favor of getting this review published.

Suggestion: Need to improve resolution of Figure 7.

Reviewer 4 Report

The review “Oligonucleotide-Recognizing Topoisomerase Inhibitors (OTIs): Precision Gene Editors for Neurodegenerative Diseases?” by Bax et al. provides overview of gene editing strategies utilizing DNA-cleavage stabilizing topoisomerase inhibitors linked to specific oligonucleotide-recognizing elements. The authors first summarize the neurodegenerative diseases which rise due to specific gene mutations and the ongoing gene therapies to intervene them. Then DNA topoisomerases, topoisomerase targeting anti-cancer drugs and selected OTIs are introduced, followed by discussion on the advantages and disadvantages of OTIs compare to other gene editing strategies.

Overall the manuscript is well written. However, before acceptance some minor corrections are needed.

Comments and Suggestions:

Lines 71-72: The comment “(proteins such as SMN are not italicized whereas corresponding genes, such as SMN1 are in italics)” can be omitted.

Lines 149 and 264: Should read “…are numbered relative to the two…” and “…fusion truncate is shown in…”

Line 288: Reference #.34 is Arimondo et al., not Infante Lara et al.

Lines 316-315: The sentence “[ Then enhancing the stability of the OTI-DNA target strand complex (relative to the DNA-DNA homoduplex) should encourage strand invasion. ]” seems to be redundant.

Line 354: Figure 6 can be deleted. The paragraph “lines 343-353” would be sufficient.

Lines 364-365: CRISPR/Cas9 gene editing is used also in lower eukaryotes, such as S. cerevisiae or S. pombe. The authors might add the relevant references here.

Line 471: Shouldn’t this be as “Can OTIs combine the powers of CRISPR/Cas and ASOs and lack their inherent weaknesses as in vivo gene editors?”, without a word “Discussion” ? Please, check and correct if necessary.

Lines 476-478: The statement “However, obtaining OTIs with specificity for bacterial over human topoisomerases and how to deliver such OTIs to bacteria may be two of the reasons for our lack of success in obtaining funding for such research to date” sounds as personal complain and authors might consider to delete or reformulate this.

Round 2

Reviewer 1 Report

I thank for the authors for their considerate and comprehensive response to my comments, and I am very satisfied with the changes to the manuscript. Therefore, I can now recommend this manuscript for publication.

Author Response

We would like to thank this reviewer for their comments and are pleased that they now say.

' I thank for the authors for their considerate and comprehensive response to my comments, and I am very satisfied with the changes to the manuscript. Therefore, I can now recommend this manuscript for publication.'

Reviewer 2 Report

This paper is qualified to be published on IJMS.

Author Response

We thank the reviewer and are pleased to see they now say:

'This paper is qualified to be published on IJMS.'